# Algorithms and Resources for the Monitoring of Very-Low-Frequency Signal Deviations Due to Solar Activity Using a Web-Based Software-Defined Radio-Distributed Network

**DOI:** 10.3390/s24144596

**Published:** 2024-07-16

**Authors:** Ilia Iliev, Kostadin Tudjarov, Ivaylo Nachev, Peter Z. Petkov, Yuliyan Velchev, Ana Ilieva

**Affiliations:** Faculty of Telecommunications, Technical University of Sofia, 1000 Sofia, Bulgaria; igiliev@tu-sofia.bg (I.I.); ktudjarov@gmail.com (K.T.); pjpetkov@tu-sofia.bg (P.Z.P.); julian_s_velchev@tu-sofia.bg (Y.V.); aniilieva@abv.bg (A.I.)

**Keywords:** D-region, ionosphere, VLF, solar flare, SDR, radio astronomy, SID, monitoring system, solar activity monitoring, software-defined radio, very-low-frequency signals

## Abstract

This work presents the development and testing of an experimental web-based SDR (software-defined radio) monitoring system for indirect solar activity detection, which has the ability to estimate and potentially predict various events in space and on earth, including solar flares, coronal mass ejections, and geomagnetic storms. The proposed system can be used to investigate the effect of solar activity on the propagation of very-low-frequency (VLF) signals. The advantages and benefits of the given approach are as follows: increasing measurement accuracy and eventual solar activity identification by combining measurements from multiple spatially distributed SDRs. The verification process involves carrying out several experiments comparing data from the GOES satellite system and the Dunksin SuperSID system with information received by the SDR monitoring system. Then, utilizing Pearson correlation coefficients, the measured data from the SDRs, along with those from the GOES satellite system and the Dunsing monitoring station, are investigated. At the time of a solar flare, the correlation value is above 90% for most of the stations used. Combining the signal-to-noise ratio via summation also shows an improvement in the results, with a correlation above 98%.

## 1. Introduction

The monitoring of solar activity and knowledge of solar processes are fundamental to human activity. The sun is the main source of energy in our solar system; for this reason, in radio astronomy, the Sun determines the conditions in the Near-Earth space (so-called Space weather). By studying the activity of the sun, the scientific community can explore and create tools to predict various events in space and on earth, including solar flares, coronal mass ejections, and geomagnetic storms.

Space weather forecasts are strategically important for the protection of various systems affected by solar activity. These include the following [1,2]:−Power infrastructure: Intense solar activity can cause geomagnetically induced currents in power grids that can disrupt the stability and reliability of these systems and networks. The proactive monitoring of solar activity helps power grid operators to anticipate and manage the effects of geomagnetic storms.−Radio communications: Solar flares and other solar emissions can affect radio communications, both terrestrial- and space-based, including aviation, marine, emergency services, etc. Solar activity monitoring is essential for adapting the radio communication channels of radio systems to maintain reliable communication.−Human health: Radiation from the sun poses a potential threat to astronauts, exposing them to increased radiation levels. Geomagnetic storms can also affect human health, including neurological processes and mood. Observations of solar activity can provide information that can define measures for protecting astronauts and human health in space.−Spacecraft: Observations of the Sun allow for accurate models and predictions of events in space, supporting effective management and actions to prevent possible adverse effects on spacecraft and astronauts.

In summary, monitoring solar activity is critical for predicting space weather, protecting satellites and infrastructure, ensuring reliable communications, protecting astronauts and spacecraft, and advancing our understanding of solar processes and their impact on Earth and beyond. Different tools and systems are used to study and characterize the different features of solar activity, such as [3,4] the following:(a)Solar telescopes: used for high-resolution, direct observations of the Sun;(b)Coronal telescopes: this type of telescope focuses on the sun’s coronal region, where high-temperature activity occurs;(c)Radio interferometers: analyze emissions in the radio range of the sun’s electromagnetic radiation;(d)Magnetometers: measure the parameters of the sun’s magnetic field that are related to solar activity;(e)Observations of ionospheric dynamics using VLF radio waves: this method indirectly studies changes in the ionosphere due to solar activity. Changes in the ionosphere resulting from solar flares are determined and analyzed by studying very-low-frequency (VLF) radio wave propagation in the ionosphere. The Global VLF Monitoring Program project uses this approach to study solar activity. These can be ground-based or satellite-based systems, and they work cooperatively or individually to provide a complex picture of solar activity, allowing for the prediction of solar processes and the investigation of solar physics.

Solar flares are characterized by a rapid and intense change in star emissions; the condition for their occurrence is a release of magnetic energy stored in the solar atmosphere. A solar flare emits electromagnetic waves across the spectrum, from very long wavelengths, through microwaves, to the high-frequency regions of X-rays and gamma rays [5]. In general, solar flares are measured in watts per square meter—${W/m}^{2},$ and they are classified as A (${{<10}^{-7}\;W/m}^{2}$), B (${10^{-7}{÷10}^{-6}\;W/m}^{2}$), C (${10^{-6}{÷10}^{-5}\;W/m}^{2}$), M (${10^{-5}{÷10}^{-4}\;W/m}^{2}$), or X (${{>10}^{-4}W/m}^{2}$). Each X-ray class category is ordered along a logarithmic scale from 1 to 9, e.g., B1 to B9, C1 to C9, etc. An X2 flare is twice as powerful as an X1-type flare and is four times more powerful than an M5 flare. The category of X-class eruptions is slightly different and does not end at X9 but continues up. Solar flares of X10 or higher are also sometimes called super X-class solar flares [5,6].

The research approach employed in this area is based on the ionospheric density variation in charged particles as a result of a solar eruption. The results show that it is possible to determine the occurrence and give an estimate of a solar flare based on ionospheric density data. This effect is also associated with a decrease in the height of the lowest layer of the ionosphere. Monitoring these kinds of variations is therefore of great importance for studies of VLF wave propagation into the ionosphere. However, ion propagation in the ionosphere also depends on several additional atmospheric meteorological factors, which introduce some uncertainty in the prediction of solar flares.

On the other hand, electromagnetic waves in the VLF range of 3–30 kHz propagate thousands of kilometers in the waveguide formed between the earth’s surface and the ionosphere [7]. During the day, the ionospheric D-layer (at an altitude of 60–90 km) appears as the upper waveguide boundary; meanwhile, at night, the D-layer has a low concentration of ions, and then reflection takes place from the higher layers. Diurnal and seasonal changes in the ionospheric concentration lead to predictable changes in reflection height, while space weather events lead to random changes in the ionosphere. Naturally, these variations also lead to variations in the propagation of VLF electromagnetic waves. This phenomenon, as well as their reflectivity from the D-layer of the ionosphere, makes them a suitable tool for indirect studies of solar activity. The direct measurement of ion concentration in the D-layer is impractical because its height is too low for satellites and too high for weather balloons. The ability of these waves to probe these parts of the atmosphere is therefore highly important. The difficulty comes from studying the natural propagation of VLF in a difficult medium such as plasma, which is the medium of the ionosphere [8].

The concentration of current carriers in the ionosphere and the permeability of electromagnetic waves in the Q-band have been measured and analyzed for many years using measurements from ground-based sounding stations [7].

Many powerful VLF transmitters are located in various locations around the world. The VLF electromagnetic waves they emit can be used to identify changes in the ionosphere caused by solar activity [8,9,10]. This is achieved using receiving radios that measure the power and/or received VLF signal phase. By analyzing the changes in the received signal parameters, solar activity can be identified. For example, in [11], the characteristics of and differences in the responses of the measured parameters of the received VLF signal caused by solar flares are investigated by comparing the responses of the VLF signal to solar flares during the day and at night. This approach requires a dedicated measurement receiver placed at a suitable location with connectivity to a monitoring system that accumulates and stores the measured powers and/or phases of the received signal. A dedicated monitoring station must be established and maintained if this approach is selected, which requires additional resources.

The software-defined radio (SDR), with its current and future applications in relation to cognitive and conventional radio networks with artificial intelligence, is a suitable tool for monitoring electromagnetic radiation artificially created by human activities or for studying physical natural phenomena, such as solar activity. Numerous ground-based SDR stations for amateur radio and other purposes have emerged and are being built in multiple locations around the world, offering free access via a WEB interface. One of the most widespread systems is KiWiSDR [10]. KiWiSDR is a versatile tool that has applications in a variety of fields, including amateur radio activities, education, spectrum monitoring, signal analysis, and research. It allows enthusiasts, students, and professionals to explore and experiment with radio signals without the need for specialized hardware and without the need to be physically present near the receiver itself.

The proposed technical solution aims to leverage terrestrial spatially distributed radio receivers, utilizing existing open-access WEB-based software-defined radios (SDRs), for the purpose of analyzing and detecting solar flares. This entails measuring and processing received signal power data from high-power stationary very-low-frequency (VLF) transmitters. The envisioned approach offers several strong advantages that are easy to demonstrate, as follows:(1)It eliminates the necessity of establishing and operating dedicated monitoring stations, thereby streamlining the deployment process and reducing the associated costs. Additionally, it enables the collection and analysis of data not only from a single receiver but from multiple SDR receivers that are spatially dispersed across the target’s desired geographical area. This spatial distribution enhances the robustness and reliability of the observations. It also secures the data supply, as disturbances or defects in one station leave many others functioning and submitting data, so the overall system remains operational.(2)Amalgamating and processing the results obtained from distributed reception creates an increased potential for more precise identification of solar activity through the application of sophisticated algorithms. This amalgamation facilitates a comprehensive understanding of solar flare dynamics and their impacts.(3)The proposed system allows for concurrent operations with multiple stationary VLF transmitters worldwide, broadening the scope and scale of the observations. This global reach enhances the comprehensiveness, universality, and depth of the data collected, contributing to a more comprehensive understanding of solar phenomena.

In summary, the utilization of terrestrial spatially distributed radio receivers, coupled with open-access WEB-based SDR technology, represents a promising avenue for advancing the detection and analysis of solar flares. By capitalizing on existing infrastructure and facilitating collaborative data collection and analysis, this approach holds significant potential for enhancing our understanding of solar activity and its implications.

## 2. The Proposed System: Hardware and Working Principles

### 2.1. Single-Sensor Monitoring System

To conduct an indirect study of solar activity by measuring the power and/or phase of the VLF radio signal reflected from the ionosphere, a single monitoring station (receiver) with the ability to measure the radio signal parameters and record the results, as well as an appropriate means to analyze these results, is required. Many systems of this type have been developed and implemented [8,9,11]. An example block diagram of a single-sensor monitoring system is shown in Figure 1.

By varying the frequency of the receiver, the power of signals emitted and reflected by different VLF transmitters can be measured. In this case, however, this type of measurement is limited to monitoring only the reflected signals emitted from a single transmitter. Another problem of a single monitoring system is that the measured results are related to the geographical location of the monitoring station and the monitored transmitter. Using this measurement concept, the apparently diurnal variation in the received signal parameters strongly depends on the above conditions. In this case, there is a possibility that changes in the monitored signal parameters that carry information about solar activity may be missed [12].

The above barriers to increasing the reliability of solar activity identification can be overcome with the use of a spatially distributed multi-sensor system composed of multiple local stations. To achieve this goal, single monitoring systems can be combined into a common network where the measured information is collected and processed centrally. Here, we propose a spatially distributed monitoring system that does not use dedicated monitoring stations for solar activity monitoring but rather uses territorially distributed SDR receivers used for general use, with remote access capabilities and corresponding functionalities to measure the VLF signal parameters. KiWiSDRs positioned around the globe with open access are suitable for this purpose. A current list of active KiWiSDRs can be found at http://rx.linkfanel.net/ (accessed on 5 July 2024).

### 2.2. Multi-Sensor Monitoring System

This system relies on the influence of solar activity on the propagation of very-low-frequency (VLF) transmitter signals. By analyzing these signals, which are received by various spatially distributed and Internet-connected software-defined SDR receivers, the level of solar activity can be assessed. The system structure and operational concept are shown in Figure 2. At the core of the system is a KiWiSDR software-defined radio equipped with a web-based API (application programming interface) and interface, serving as the primary node.

### 2.3. KiWiSDR

KiWiSDR (V.1) [10,12] is a software-defined radio platform that enables users to access and control radios remotely via the Internet. The platform is a means to explore the radio spectrum using a Web browser. The operating frequency range of the devices is typically 10–30 MHz (VLF/LF/MW/HF).

Users can access KiWiSDR receivers from any location worldwide using a web browser. The web interface offers a plethora of features, enabling users to tune the receiver to various frequencies, adjust the receiver bandwidth, employ different methods of analog and discrete demodulation, and decode information transmitted across diverse communication and navigation systems. Figure 3 shows a Kiwi wideband shortwave radio receiver map [13].

### 2.4. Software Implementation

Based on the open interface of KiWiSDR, software was created to manage the SDR receiver by organizing a web-socket-type connection. All commands and measurement data are transferred through this interface.

The software implementation includes the development of web-based SDR measurements of the received power and recording the received power from a few selected receiver systems for management. For this purpose, the task is divided into two parts related to the implementation of different system functions as follows:−The process involves establishing a connection to the KiWiSDR organization website, issuing specific commands to control and configure the SDR, and subsequently processing and recording the received information;−The organization and management of individual solar activity surveys, with the ability to select KiWiSDR receivers and set the main survey parameters.

In developing the system to solve the two parts of the problem, two separate software modules were implemented, conventionally named the “module for connecting to KiWiSDR, processing and recording the information obtained” and the “module for organizing and managing research”.

The modules are implemented with Python 3.11.3, using PyCharm.

### 2.5. Module for Connecting to KiWiSDR and Processing and Recording the Received Information

In this study, the program designed by Olaf (LA3RK) [14] for coupling KiWiSDR, reception, discrete signal report recording, and signal-to-noise ratio (SNR) estimation served as the foundation. The accompanying description provided within the snrtorrd.py file [14] indicates that the data are stored in a round-robin database and that the (?) serigraph is utilized for plotting, neither of which was adopted in the development of the current system. Our approach utilizes solely the following: (a)The socket communication element of the KiWiSDR server, with minor modifications aimed at enhancing module stability;(b)The processing and computations applied to the acquired data. We propose an original solution for creating, storing, and updating the information received from each SDR receiver in four CSV (comma-separated values) files as follows:
○With average values, the file name is given by the user and the extension “.csv”;○With peak power values, with the name given by the user and the add-on “_peak.csv”;○Power in dBm, with the name given by the user and the addendum “_dBm.csv”;○With the values of SNR, 95th percentile, and median, with a name given by the user and an addendum “_snr.csv”.

In the event of a failed connection to the KiWiSDR server or incomplete data receipt, the methodology is expanded to include the insertion of a dummy entry, denoted as “−999”, not only in the file containing the average values but also in all other pertinent system files.

### 2.6. Research Organization and Management Module

In order to facilitate studies of solar activity, a program module, named StartPROJECT.exe was developed. This module offers users the ability to select and configure various parameters, including the operating frequency, the duration of intervals for data exchange with the respective server, and the time interval for conducting studies, as well as the addresses, ports, and corresponding file names for storing information from each monitored KiWiSDR server.

The module is designed to facilitate solar activity surveys by initiating connections to KiWiSDR at predefined intervals, processing and recording the received data, and orchestrating surveys for particular frequencies utilizing three KiWiSDR receivers simultaneously.

Using the Tkinter module, the graphical user interface was created. Python has and can use many GUI modules, but Tkinter is the only one that is built into the Python standard library. Tkinter has several strengths. It is a cross-platform module, so the same code runs on Windows, macOS, and Linux. Graphical elements are rendered using built-in operating system elements, so applications created with Tkinter look as if they belong to the platform on which they run. It therefore constitutes a compelling choice for building GUI applications in Python.

The Tkinter elements (widgets) that were used to create the application include the following:−Label—an element used to display text on the screen;−Entry—a text entry element that allows for the entry of only one line of text;−Button—a button-type element that can contain text and can perform a click action.

Figure 4 and Figure 5 show the working algorithm of the module for connecting to KiWiSDR and processing and recording the received information.

The application layout in Tkinter is controlled by geometry managers. Tkinter has the following geometry managers: “.pack()”, “.place()”, and “.grid()”. “.grid()” is used in the proposed monitoring system; it works by dividing a window or frame into rows and columns. The element location is determined by calling .grid() and specifying the row and column indices to the row and column keyword arguments, respectively.

The development of the module also used the appropriate tools for changing the content of the input elements and blocking/enabling buttons. Table 1 presents all of the necessary input parameters of the described system.

When writing the information from the SDR server, “.csv”, “_peak.csv”, “_dBm.csv”, and “_snr.csv” are added to the file name.

## 3. Test and Measurement Results

The solar activity detection results with the implemented SDR monitoring system are analyzed in this section. The data collected using the developed system are compared with published and measured results concerning solar activity, which are based on measurements using detection technology (such as public data from SpaceWatherlive), and relevant conclusions are drawn concerning the feasibility of development.

Solar flares can significantly affect the received signal power and the signal-to-noise ratio (SNR) of propagating VLF radio signals through ionosphere reflection. In the summer of 2023 (starting from 27 June 2023), this experimental SDR monitoring system was tested using different KiWiSDRs at launch. The results of the conducted system performance tests are presented below, using samples from the period 25 July 2023–9 August 2023. The resultant data are compared with solar activity data published on the online platform SpaceWeatherLive (https://www.spaceweatherlive.com/) (accessed on 5 July 2024).

A 23.4 kHz VLF transmitter, DHO38 Rhauderfehn, Germany, 53.087341° N–7.608652° E, was selected for the verification studies and the operation of the proposed system. The DHO38 VLF transmitter has been used for communication with the submarine fleets of NATO countries and has an output power of approximately 800 kW.

As described above, the system testing employed software-defined radios (SDRs) operating at a frequency of 23.4 kHz (see Figure 6), which are situated at various geographical locations, as follows:−Esztergom (Grante Tower), Hungary, http://194.36.180.89:8073/ (accessed on 5 July 2024), at a distance of 985 km from the transmitter;−Bjargtangar (Westfjords), Iceland, http://tangar.utvarp.com:8073/ (accessed on 5 July 2024), 2253 km from the transmitter;−Malmefjorden, Norway, http://la1plahf.zapto.org:23462 (accessed on 5 July 2024), at a distance of 1084 km from the transmitter;−Vejby, Denmark, http://oz1bfm.proxy.kiwisdr.com:8073 (accessed on 5 July 2024), at a distance of 444 km from the transmitter, SDR, etc.

The information proposed by NOAA (https://www.swpc.noaa.gov/) (accessed on 5 July 2024) shows that, in the selected period of 26 July 2023 to 5 August 2023, geomagnetic storms G1, G2, and G3 occurred. This is proof, based on actual confirmed data, that the system can be used to identify such events.

From Figure 7, we can clearly see the increased solar activity manifested by the M4.63 flare, which started at 10:17 a.m., with a maximum at 10:37 a.m., and ended at 10:48 a.m. Here, the levels R1–R5 are associated with the NOAA five-level rating scale, indicating the severity of an X-ray related to a radio blackout.

Figure 8 and Figure 9 show the power variations in dBm and SNR recorded by the system on 26.07.2023 from Esztergom (Grante Tower); Bjargtangar (Westfjords); Malmefjorden; Vejby; the GOES satellite system [15]; and the Dunksin SuperSID system [16,17].

The received power levels from the chosen receivers have different average values because of their locations at different distances from the VLF transmitter and system. The received signals from the Iceland KiWiSDR have the highest power, more than those in Denmark. For comparison, in the lower part of Figure 8, the power density from two GOES X-ray sensors is shown. The first operates with particles from 1 to 8 angstroms and the second with particles from 0.5 to 4 angstroms. The data are from https://vlf.ap.dias.ie/data/dunsink/ (accessed on 5 July 2024). The solar flare M4.63 class is distinguishable and has a duration of 31 min. The measured results similarly show an increase in the received signal values.

Figure 9 allows us to compare the measured results of the proposed system with those obtained from the Dunsing monitoring station (DIAS, Ireland), presented as a signal-to-noise ratio (SNR). Dunsing is a DIAS observatory located near Dublin (Ireland), where, in addition to optical astronomy, indirect measurements of solar activity using the SuperSID system are performed. In the same figure, a plot of the SNR ratios obtained by the simultaneous summation of the levels from the four SDR receivers is presented. 

Based on the data presented in Figure 8 and Figure 9, it is evident that the system successfully documented fluctuations in the received signal power following the solar flare on 26 July 2023. Combining the measured power (signal-to-noise ratio) from multiple stations through summation improves the system’s noise immunity. Additionally, the stations in Iceland and Denmark reported an increase in the received signal level at 16:01, corresponding to a class M solar flare, while the Dunsing station reported no change in received power. 

Utilizing Pearson correlation coefficients, the correlation between the measured data from KiWiSDR with those from the GOES satellite system and the Dunsing monitoring station was investigated. The correlation calculation was performed using a sliding window. The duration of the window was set to be about one hour, i.e., twice the duration of the solar flare at 10:36 (UTC). The correlation results are given in Figure 10a–c. Figure 10 shows the correlations (a) with GOES data for the X-ray flux from 1 to 8 angstroms; (b) with GOES data for the X-ray flux from 0.5 to 4 angstroms; and (c) with the data from the Dunsing SuperSID system. The correlation value is above 90% at the time of the solar flare at 10:36 (UTS) for most of the KiWiSDR stations. The correlation coefficients at the time of the solar flare at 10:36 (UTS) are presented in Table 2.

The data from the figures and table show a high correlation value at the time of the solar flare. The correlation value is higher when the signal-to-noise ratio of the measured signal is high. The Norway station’s data show a typical correlation value below 90%. Combining the signal-to-noise ratio by summation also shows an improvement in the result, with a correlation above 98%.

One problem with this monitoring approach is that certain notable challenges occur in the form of sporadic failed connections to KiWiSDR servers or incomplete data reception. When selecting the proper KiWiSDR station, it is necessary to choose one that not only has the required measurement frequency range but also a suitable antenna system, which is optimal for the reception of radio signals in the VLF band. The results shown above demonstrate the feasibility of this approach with spatially distributed monitoring stations and a distributed reception, which leads to higher-quality measurements of the parameters of the received signals, as well as higher accuracy in solar activity identification.

## 4. Discussion and Conclusions

The proposed approach transforms a widely deployed amateur radio system into a dedicated and powerful tool for researching and predicting how solar activity in the ionosphere affects the propagation of VLF radio waves, a topic that is highly important for human activity, industry, ecology, and earth research.

The proposed system for monitoring ionospheric disturbances in the D-layer caused by solar flares through the observation of VLF signals exhibited promising outcomes. These results were acquired through the surveillance of ionospheric variation across Western Europe, indicating the system’s efficacy for the prolonged monitoring of such solar phenomena. This finding was confirmed through the validation of the measured outcomes and a comparative analysis of the information received by the SDRs monitoring system with data from the GOES satellite system and from the Dunksin SuperSID system. The correlation value was above 90% at the time of a solar flare for most of the examined stations.

The technical solution developed underscores the feasibility of deploying not just one but multiple ground-based radio receiving stations, strategically dispersed across space. This multi-station monitoring system, designed for analyzing and detecting solar activity by measuring and processing the received signal power from potent stationary VLF transmitters, is readily implementable owing to its utilization of existing, freely accessible web-based SDRs.

The advantages and benefits of the proposed approach are as follows: increasing measurement accuracy and eventual solar activity identification by combining measurements from multiple spatially distributed monitoring stations; improving the monitoring system’s reliability by using multiple interchangeable SDR-based receivers; and the easy, fast, and cheap implementation of a monitoring system via implemented KiWiSDR platforms. Combining the signal-to-noise ratio by summation also shows an improvement in the result, with a correlation above 98%.

One notable challenge concerns occasional failures to connect to the KiWiSDR server; meanwhile, partial data loss may occur. This risk could be mitigated by expanding the network of SDRs involved in the monitoring process.

Future follow-up work will focus on research methodologies and tools for processing the accumulated data, with an emphasis on improving techniques for monitoring the propagation of VLF signals in the ionosphere and their modification due to increased solar activity. The monitoring of solar activity using only VLF data should be considered with caution. For a better study of solar activity, the measured data from the considered system can be combined with solar activity data from other systems, which will increase the resolution for detecting and studying ionospheric variation.

## Figures and Tables

**Figure 1 sensors-24-04596-f001:**
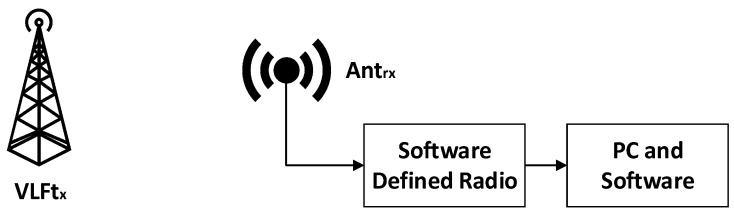
Block diagram of a single-sensor monitoring system.

**Figure 2 sensors-24-04596-f002:**
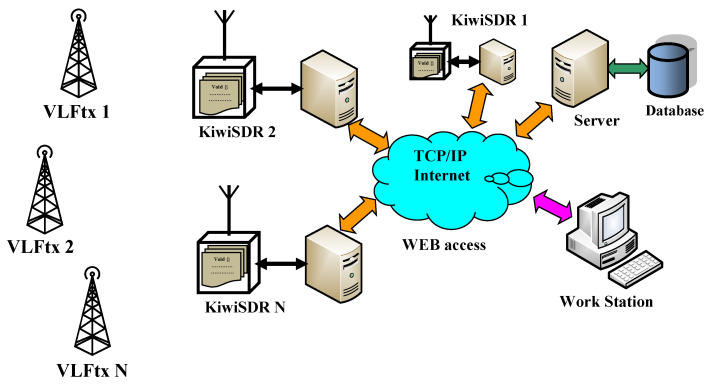
Block diagram of a realized and tested system for monitoring ionosphere activity by means of VLF radio waves.

**Figure 3 sensors-24-04596-f003:**
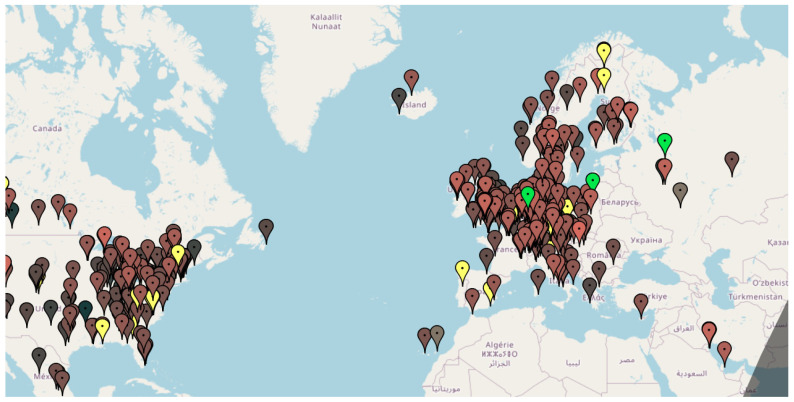
Kiwi wideband shortwave radio receiver map.

**Figure 4 sensors-24-04596-f004:**
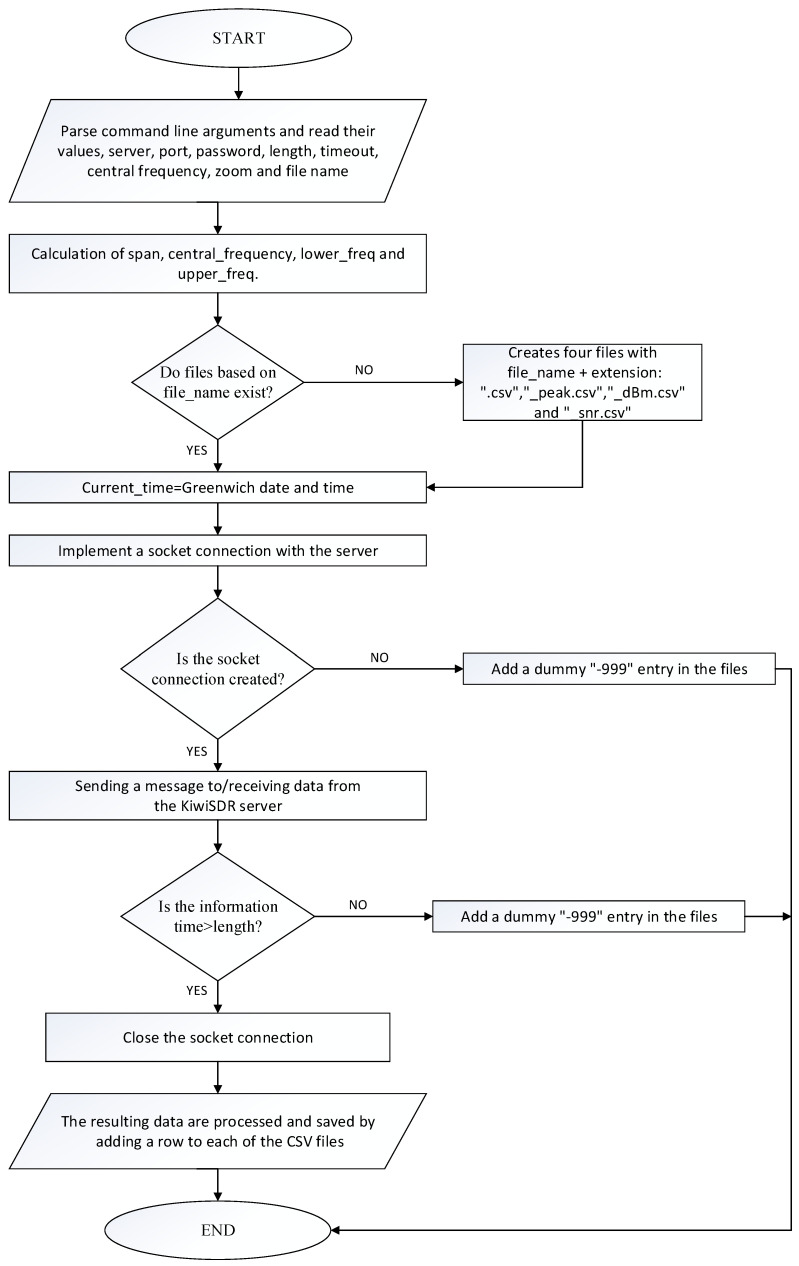
The algorithm used to connect to the KiWiSDR server and to process and record the received information.

**Figure 5 sensors-24-04596-f005:**
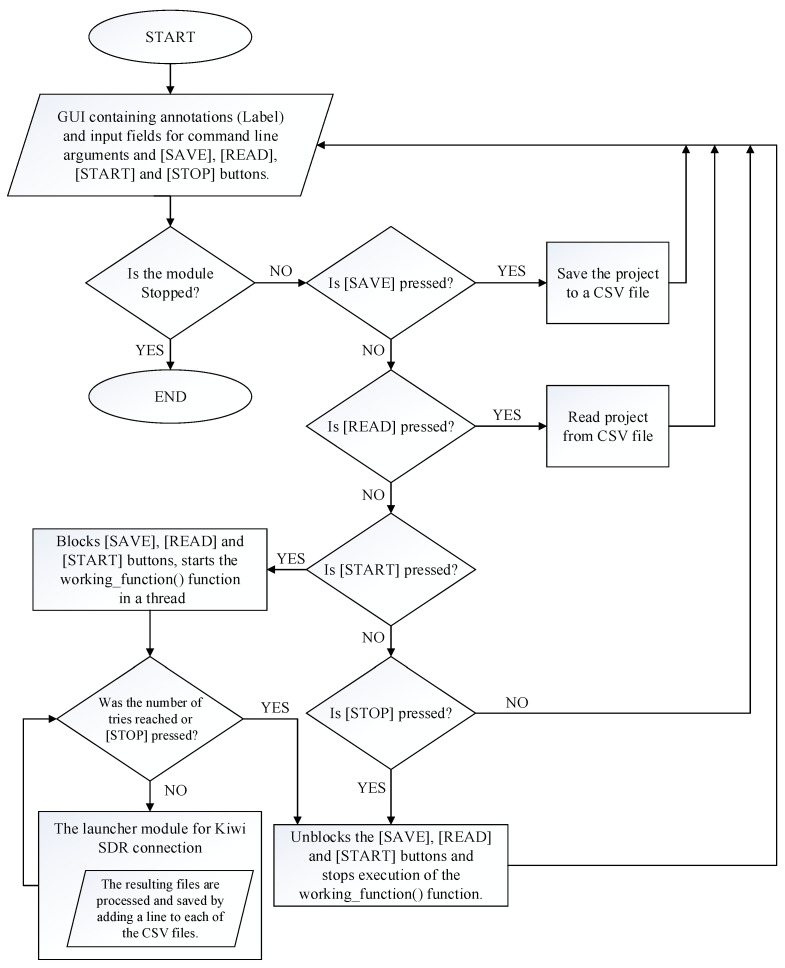
The algorithm of the research organization and management module.

**Figure 6 sensors-24-04596-f006:**
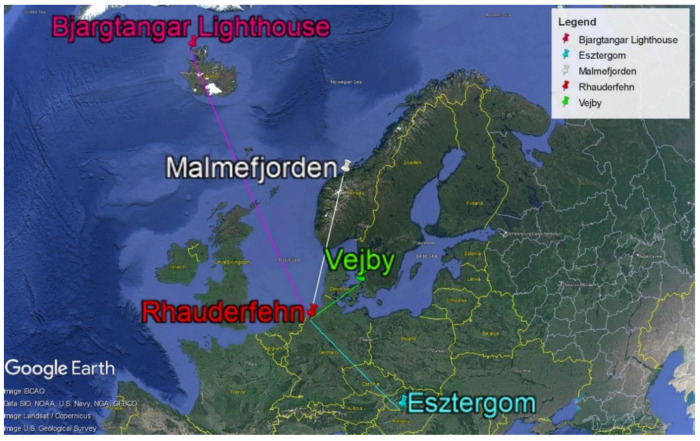
Locations of the transmitters and used SDRs.

**Figure 7 sensors-24-04596-f007:**
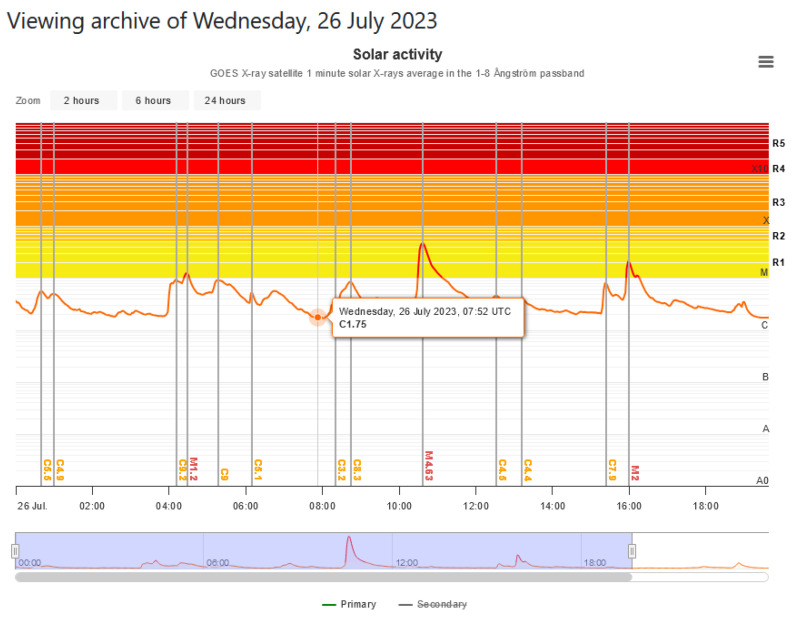
Solar activity from 26 July 2023 (SpaceWeatherLive).

**Figure 8 sensors-24-04596-f008:**
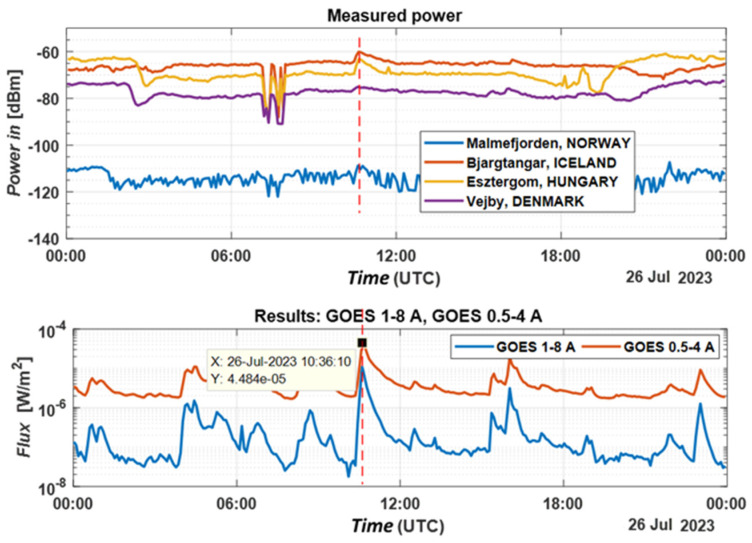
Power data in dBm recorded by the system on 26 July 2023.

**Figure 9 sensors-24-04596-f009:**
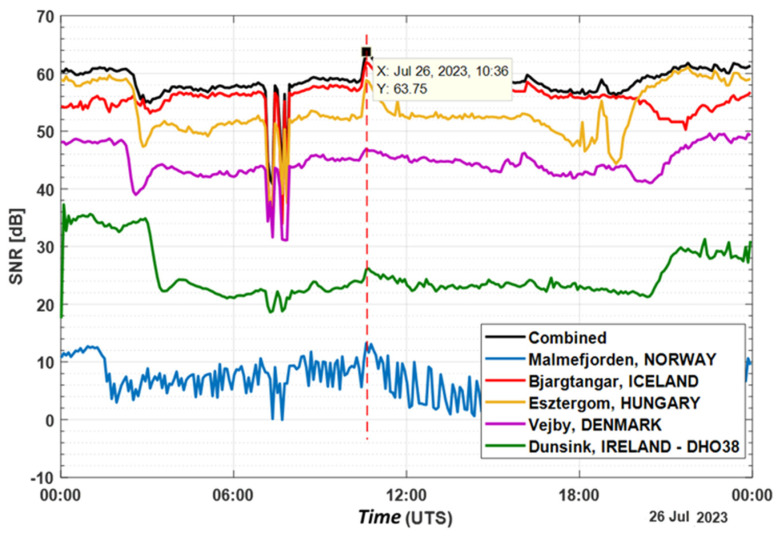
SNR data recorded by the system on 26 July 2023.

**Figure 10 sensors-24-04596-f010:**
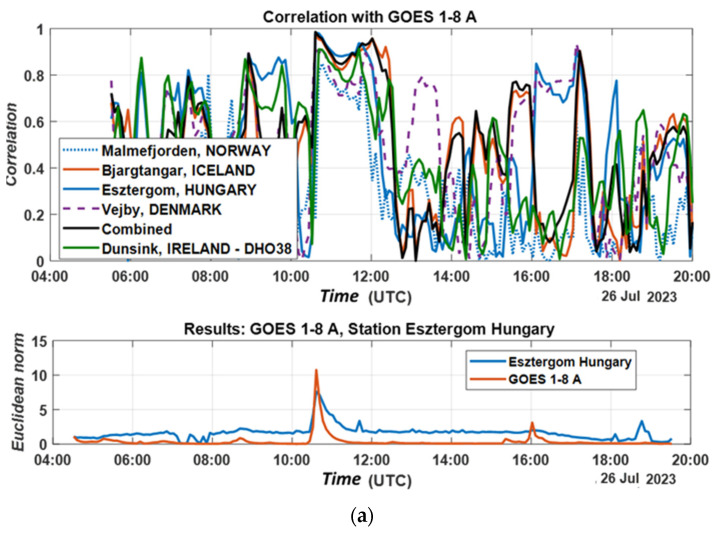
(**a**–**c**) Correlations for solar activity on 26 July 2023.

**Table 1 sensors-24-04596-t001:** Input system parameters.

Designation	Value	Explanatory Note
proj_file	t	CSV project file name
$f_{c}$	23.4	Used frequency [kHz]
Interval	300	Investigation interval [s]
Attempts	10,000	Number of studies
Timeout	60	Connection waiting time [s]
SDR1	194.36.180.89	Server IP/DNS address
Port1	8073	Number of port
File1	Esztergom_HUNGARY	File name (common part)
SDR2	db0bbb.dnshome.de	Server IP/DNS address
Port2	8073	Number of port
File2	Bernau_Bei_Berlin_	File name (common part)
SDR3	wessex.zapto.org	Server IP/DNS address
Port3	8073	Number of port
File3	Wessex_ENGLAND	File name (common part)

**Table 2 sensors-24-04596-t002:** Maximum correlation coefficients for solar activity on 26 July 2023 at 10:36 (UTS).

Stations	Correlation with GOES 1–8 Å	Correlation with GOES 0.5–4 Å	Dunsing, Ireland
Esztergom, Hungary	0.981	0.969	0.9751
Bjargtangar, Iceland	0.9714	0.9787	0.9578
Malmefjorden, Norway	0.8108	0.8015	0.879
Vejby, Denmark	0.9285	0.933	0.8863
Combined	0.9861	0.9874	0.9652
Dunsing, Irland	0.911	0.9105	-

## Data Availability

Data available in a publicly accessible repository that does not issue DOIs: http://kiwisdr.com/public/ (accessed on 5 July 2024) and https://vlf.ap.dias.ie/ (accessed on 5 July 2024).

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
