# Peer review of "Algorithms and Resources for the Monitoring of Very-Low-Frequency Signal Deviations Due to Solar Activity Using a Web-Based Software-Defined Radio-Distributed Network"

_sensors, 2024, doi:10.3390/s24144596_

Round 1

Reviewer 1 Report

Comments and Suggestions for Authors

In the manuscript, a system for detection VLF signals with the use of many receivers is proposed. This approach is hopeful for reliable identification of ionospheric disturbances. I suppose the idea of this study is interesting and potentially fruitful.

At the same time, the current version of the manuscript has noticeable flaws. Mainly, the Discussion section and the review in the Introduction are insufficient because various sources impacting on VLF signal propagation are not discussed. Solar activity is not the same as geomagnetic activity, and the ionosphere is dependent on processes not only ‘from above’, but ‘from below’ as well. As a result, monitoring of solar activity by only VLF data should be considered with caution, and sentences about forecasting are not reliable in this case. The list of references should be expanded, now it includes seven scientific papers (one of them is mentioned three times) and four Internet sites.

Line-by-line comments are included in the attached file. Taking into account all the details, I have proposed major revision at this stage.

Comments on the Quality of English Language

The manuscript contains few significant grammatical errors, but there are rare wrong wordforms or unclear phrases. All this is not critical. Further attention should be paid to the structure of the manuscript because it currently includes three subsections numbered as 2.2. The list of references should be checked in addition to its expanding.

Author Response

The recommendations made by the reviewer is sound and all objections made have been removed. We believe that thanks to them the work acquired the necessary quality and clarity.

L. 2. ‘solar activity detection’ – in my opinion, the paper is not devoted directly to solar activity, but to the VLF

Author’s reply - the title has been edited for better understanding

L. 4. All authors are affiliated with Faculty of Telecommunications, Technical University of Sofia. In my opinion, no need is to insert designations 1–6 for affiliation in this case.

Author’s reply - the template requirement is fulfilled

L. 33-34. It is worth including only one term from ‘VLF’ and ‘VLF signals’.

Author’s reply - as VLF keywords also include VLF signals

L. 39. ‘the Sun determines cosmic time’ – it is unclear what is understood as ‘cosmic time’, the phrase should beedited.

Author’s reply - hass been change with ‘the Sun determines the Sun determines space weather’

L. 81. ‘Solar flares are characterized by a rapid and intense change in the star brightness’ – this statement should Authors’replybe ckarified. The brightness of the Sun in white light changes rarely and weakly (otherwise, life on the Earth would be very problematic).

Author’s reply - hass been change with "Solar flares are characterized by a rapid and intense change in the star emissions, the condition for their occurrence is the release of magnetic energy stored in the solar atmosphere."

L. 86. ‘W/m2’ – it is worth using superscript, ‘W/m2’.

Author’s reply - fixed according to the recommendation

L. 88. ‘An X2-type flare is twice as powerful as an X1-type flare and four times as powerful as an M5 flare’ – a more detailed description of the classification would be useful.

Author’s reply - in our opinion, it would repeat some information in the mentioned references

L. 89. ‘and does end at X9’ – rather ‘and does not end at X9’.

Author’s reply - fixed according to the recommendation

L. 175. ‘an single seonsor’ – should be ‘a single sensor’.

Author’s reply - fixed according to the recommendation

L. 203. ‘The system structure and operational concept are shown on Figure 1 below’ – should be ‘in Figure 2 below’.

Author’s reply - fixed according to the recommendation

L. 208. ‘2.2. KiWi SDR’ – the numbering of subsections should be corrected (see L. 198, L. 219).

Author’s reply - fixed according to the recommendation.

L. 218. Some comments are desirable about Fig. 3a. In addition, Fig. 3 should be cited in the text.

Author’s reply - description of relevant figures added

L. 236. ‘[19]’ – the list of references on P. 16 includes only 13 references.

Author’s reply - all orderings and reference numbers have been reviewed in detail 1 more time and have been edited and revised

L. 243. ‘Olaf - LA3RK [13]’ – some comments are needed because the paper of Gu et al. (2023) does not mention ‘Olaf – LA3RK’ in its text. This remark also applies to a file ‘snrtorrd.py’ (L. 245).

Author’s reply - due to a technical error the wrong quotation has been inserted, the appropriate source has been added in the text

L. 265. ‘Figures 4 and 5 shows’ – the verb should be in plural, ‘show’.

Author’s reply - fixed according to the recommendation.

L. 269-270. ‘This section an author’s-designed program module, named StartPROJECT.exe, has been developed’

Author’s reply -  the grammatical structure should be improved in this sentence. -fixed according to the recommendation.

L. 279. ‘Using the Tkinter module, the graphical user interface created’ – probably ‘the graphical user interface was created’.

Author’s reply - fixed according to the recommendation.

L. 313. It is worth placing Table 1 closer to its mention in the text (L. 347) the recommendations from the other reviewers have been taken into account here by adding a reference to Table 1 before the table itself on the L. 307

Author’s reply - fixed according to the recommendation.

L. 317. ‘3. Test and measurement results:’ – the colon is not needed.

Author’s reply - fixed according to the recommendation.

L. 345. The map includes an unnecessary yellow line between Ukraine and Crimea.

Author’s reply - A google open source covenant map was used, the commented line is not the subject of the article's research, it is from the google platform itself from which the map was taken, the authors' case on the map is just the added lines of the radio references. However, we support the reviewer's opinion and the remark has been fixed, as recommended.

L. 352. ‘Figure 9. Solar activity in 26.07.2023’ – a more detailed caption is necessary with a description of both curves and symbols ‘R1, R2, … , R5’.

Author’s reply - fixed according to the recommendation

L. 376. ‘presented on Fig. 10’ – the typical form is ‘presented in Fig. 10’.

Author’s reply - fixed according to the recommendation.also the numbering of the figures was recalculated one more time

L. 386. ‘kiwiSDR’ – should be capitalized, ‘KiwiSDR’

Author’s reply - fixed according to the recommendation.

L. 389. ‘occured’ – the past tense should be ‘occurred’.

Author’s reply - fixed according to the recommendation.

L. 431-432. ‘For research articles with several authors, a short paragraph specifying their individual contributions must be provided. The following statements should be used’ – these sentences should be deleted.

Author’s reply - fixed according to the recommendation.

L. 402. ‘for research and prediction of space weather, solar activity, and electromagnetic wave propagation’ – prediction of space weather is not considered in the work and should be mentioned only as hypothetical. Forecasting of solar activity is still further from the results of the conducted research. Besides, other phenomena, including lithospheric and tropospheric, are considered as factors disturbing propagation of VLF signals. All these circumstances complicate even monitoring of solar activity by VLF observations, and forecasting is more challenging task. Accordingly, there is a need to expand the Introduction and Discussion sections to consider the mentioned issues.

Author’s reply - Indeed, the use of VLF signals is an indirect approach to the estimation of solar activity and its prediction. The work is based on a number of similarly developed systems that use the approach mentioned, this is discussed in sections 82-130, where references 7-11 are cited.

L. 405-406. ‘monitoring ionospheric disturbances in the D layer caused by solar flares through observation of VLF signals’ – in my opinion, the main output of the work is to improve monitoring of VLF propagation associated with ionospheric disturbances. Consequently, this point should be emphasized with less attention to solar activity.

Author’s reply - fixed according to the recommendation.

L. 428. ‘with an emphasis on refining monitoring techniques like diversity receiving whit signals combining and enhancing solar activity identification’ – it is worth editing the phrase. What are ‘whit signals’?

Author’s reply - fixed according to the recommendation.

L. 439. ‘Recov-ery’ – no hyphen is needed.

Author’s reply - fixed according to the recommendation.

L. 444-446. – the text repeats L. 438-439, being unnecessary in my opinion. -fixed according to the recommendation.

Author’s reply - for the sake of better readability of the whole, taking into account the present reviewer and the other reviewers, the whole conclusion has been revised and rewritten

Reviewer 2 Report

Comments and Suggestions for Authors

This paper describes the development of a web-based Software-Defined Radio (SDR) system for detecting solar activity, such as solar flares and geomagnetic storms. Developed in Python and tested on a Windows 10 PC, the system uses Very Low Frequency (VLF) signals to monitor solar effects. Validation involved comparing its data with NOAA data, confirming the system's effectiveness. However, the manuscript does not yet meet publication standards due to poor language quality, unclear figures, and a lack of quantitative evidence supporting the results.

comments:

  1. Line 20, “system developed in Python and compiled” brings confusion, python is not a compiling language (e.g. C++)

  2. The abstract is not informative enough, should include the major performance metrics, the result of the verification, the uncertainties, the sensitivities to determine solar activities.

  3. Section 2.2 introduced KiwiSDR, which is not created in this work, should be in the introduction and more brief.

  4. Line 236-239, not necessary.

  5. Line 287, Label?

  6. Line 279-289, no need to present in detail Tkinter, which all the details are available in the Python official document.

  7. Figure 10a, the zoom box not consistent, in the small red box, there are 2 peaks, but zoomed only have one peak

  8. Figure 10, the date time format is not readable, need to re-make the plot.

  9. Line 376, please elaborate on how evident. The cross-matching of time? Or measured delay? Or correlation between the flare flux and the VLF variation. Need to present numbers and uncertainty analysis to validate.

Comments on the Quality of English Language

Another major concern is language and expression, need to go through substantial language editing.

For example:

  1. Line 23: “Some really interesting…” too colloquial

  2. Line 50: “Solar activity monitoring is essential to adapt radio communication channels of radio systems to maintain reliable communication.” duplicated confusing phrases.

There is more...

Author Response

1

Line 20, “system developed in Python and compiled” brings confusion, python is not a compiling language (e.g. C++)

Author’s reply - "compiled" has been removed for better understanding

2

The abstract is not informative enough, should include the major performance metrics, the result of the verification, the uncertainties, the sensitivities to determine solar activities.

Author’s reply - Work focuses on the practical implementation of an approach for indirect estimation of solar activity by monitoring the ionospheric reaction upon reflection of electromagnetic waves from the VLF band. The aim is to show that it is not necessary to build independent measuring stations, but a ready-made network of SDRs can be used, with an application in another area, for the measurement and identification of effects in the ionosphere. It is true that the accuracy of this indirect method of solar flare identification is questionable, but it is a matter of accumulating multiple measurement data and performing statistical processing. This is a matter for future work which is mentioned in the conclusion of the manuscript.

3

Section 2.2 introduced KiwiSDR, which is not created in this work, should be in the introduction and more brief.

Author's reply - True, the authors of this manuscript have no claims on the creation of KiWiSDR. KiWiSDR with its web based platform is built and used mainly for HAM radio prupouses. Users have the option to use remote radio equipment in the relevant frequency bands and thus control their connections. What is new in this work is that it is proposed to use this already built system of territorially distributed radios to monitor the change in VLF wave propagation by reflection from the ionosphere. For this reason, the description of KiWiSDR is very short, but logically it is part of the proposed approach.

4

Line 236-239, not necessary.

Author’s reply - fixed, L.255-241  were changed to avoid repetition and preserve the meaning of the text

5

Line 287, Label?

Author’s reply - fixed

6

Line 279-289, no need to present in detail Tkinter, which all the details are available in the Python official document.

Author’s reply - fixed according reviews

7

Figure 10a, the zoom box not consistent, in the small red box, there are 2 peaks, but zoomed only have one peak

Author’s reply - fixed acording rewiews now it is figure 8 line 361

8

Figure 10, the date time format is not readable, need to re-make the plot.

Author’s reply - fixed according reviews, all under figures to 8 and 9 are corrected for the purpose of clear understanding

9

Line 376, please elaborate on how evident. The cross-matching of time? Or measured delay? Or correlation between the flare flux and the VLF variation. Need to present numbers and uncertainty analysis to validate.

Author’s reply - added clarification, fixed 379-394

Reviewer 3 Report

Comments and Suggestions for Authors

Review of the manuscript by Iliev et al., Algorithms and resources for solar activity detection using a 2 Web-based SDR network

The paper describes the implementation of a system that allows one to estimate the level of solar activity by the passage of VLF radio waves on several routes in Western Europe. For implementation, widely used SDR receivers were chosen as the basis.
The effect of the dependence of the conditions for the passage of radio waves on the state of the ionosphere, which depends on solar activity, has been known for a long time. Various services for tracking the passage of radio waves have long existed in different countries.
No statistical analysis was done in the work, only one example of registration is shown.

From a scientific point of view, there is nothing new in the effect of a solar flare shown in the work.

The assumptions made in the work about the possibility of predicting solar flares based on such measurements, in my opinion, have no basis. The fact is that the state of the ionosphere is being assessed, which is a consequence of the influence of solar activity at the point occupied by the Earth. Even if we somehow register all this impact, it in itself is not sufficient to describe the current activity of the Sun, much less predict its behavior in advance.

The observation method is very crude. The impact is visible only from very large flares, but variations are visible not only from such events. In reality, the influence of solar activity is visible on the given curves if directly compared with the solar activity curve.

The implementation of a receiving system on SDR is currently not new. Many instruments are built on SDR boards and are used to obtain truly scientific results. There is no analysis provided to justify the choice of a specific SDR board.

Specific notes on the text:

Line 95-96. "Monitoring ionospheric variations is therefore of great importance not only for the propagation of electromagnetic radio waves but also for protection of the environment." How does monitoring propagate radio waves and protect the environment? State it more clearly.

Lines 97-98. “...the concentration of current carriers in the ionosphere also depends on several additional atmospheric meteorological factors...” From the paragraph it can be understood that meteorological phenomena affect currents in the D-layer of the ionosphere. What is meant?

Lines 109-111. “Direct measurement of ion concentration in the D layer is very difficult because its height is too small for satellites and too large for weather balloons.” Do weather balloons and satellites fly in the D-layer of the ionosphere? At ~80 km?

Line 203, link to Fig. 1. Do you mean Fig. 2?

Figure 3 appears to be taken from the SDR manual. This is usually not welcome. I did not find a reference to this figure in the text.

Figures 4 and 5. Flowcharts and trivial programming details are not given in a scientific article. If there is novelty in this, then this is noted and given in the appendix.

The numbering of the figures is strange: after 5 there are 8.

In Figure 9 the signatures are very small.

Figures 10 and 11. Timestamp format is not valid. It does not contain any useful information and takes up a lot of space. All graphs can be given with a common time axis.

Table 1 does not contain information referenced in the text, or is another table missing?

Overall, the manuscript looks like a draft of a student report and cannot be published in this form in a scientific journal.

Comments on the Quality of English Language

There are quite a few phrases in the text, the meaning of which requires clarification, regardless of the language. Please make sure the phrase is correct in your native language before translating into English.

Author Response

Specific notes on the text:

Line 95-96. "Monitoring ionospheric variations is therefore of great importance not only for the propagation of electromagnetic radio waves but also for protection of the environment." How does monitoring propagate radio waves and protect the environment? State it more clearly.

Author’s reply - fixed, now the whole sentence, present on lines 83 and 84 has the form” Solar flares are characterized by a rapid and intense change in the star emissions, the condition for their occurrence is the release of magnetic energy stored in the solar atmosphere.”

Lines 97-98. “...the concentration of current carriers in the ionosphere also depends on several additional atmospheric meteorological factors...” From the paragraph it can be understood that meteorological phenomena affect currents in the D-layer of the ionosphere. What is meant?

Lines 109-111. “Direct measurement of ion concentration in the D layer is very difficult because its height is too small for satellites and too large for weather balloons.” Do weather balloons and satellites fly in the D-layer of the ionosphere? At ~80 km?

Author’s reply - the entire paragraph from line 105 to line 120 has been changed, including the English wording, which removes the inaccuracies mentioned above

Line 203, link to Fig. 1. Do you mean Fig. 2?

Author’s reply - fixed now on line 178-179 is clearly described what is torn to figure 1

Figure 3 appears to be taken from the SDR manual. This is usually not welcome. I did not find a reference to this figure in the text.

Author’s reply - Authors’reply -True, the authors of this manuscript have no claims on the creation of KiWiSDR. KiWiSDR with its web based platform is built and used mainly for HAM radio prupouses. Users have the option to use remote radio equipment in the relevant frequency bands and thus control their connections. What is new in this work is that it is proposed to use this already built system of territorially distributed radios to monitor the change in VLF wave propagation by reflection from the ionosphere. For this reason, the description of KiWiSDR is very short, but logically it is part of the proposed approach.

Figures 10 and 11. Timestamp format is not valid. It does not contain any useful information and takes up a lot of space. All graphs can be given with a common time axis.

Author’s reply - We believe that the algorithms are an important part of the proposed approach and it would be useful for readers of the manuscript to implement the proposed approach themselves

The numbering of the figures is strange: after 5 there are 8.

In Figure 9 the signatures are very small.

Figures 10 and 11. Timestamp format is not valid. It does not contain any useful information and takes up a lot of space. All graphs can be given with a common time axis.

Author’s reply - fixed; Since the moments of the maximum are very close to each other, displaying them in one coordinate system would lose the overview of a change on a larger scale. Regarding the notes on the corresponding figures - The sequence of figure numbering is now preserved and all figures in the publication are sequenced

Table 1 does not contain information referenced in the text, or is another table missing?

Author’s reply - fixed –added clarification  310-311

The recommendations made by the reviewers are sound and all objections made have been removed. We believe that thanks to them the work acquired the necessary quality and clarity.

Round 2

Reviewer 1 Report

Comments and Suggestions for Authors

The text was significantly revised after the first review. In particular, the description of the obtained results was expanded and Discussion and Conclusions were changed. In my opinion, the current version of the manuscript better corresponds to the main goal of the work. Now the results for VLF data are emphasized, which is justified. In addition, the list of references has been ordered and a series of technical corrections have been made.

I assume that the paper can be published after minor revision, and my remarks at this stage are included in the attached file.

Comments on the Quality of English Language

Some grammatical errors have been yet available in the text, but they can be easily corrected. In particular, several unclear phrases were noted, they are indicated together with other comments.

Author Response

L. 16. ‘solar activity detection’ – possibly ‘indirect solar activity detection’.
Autors Response:we accept the correction, now ‘solar activity detection’ is replaced by ‘indirect solar activity detection’; now is line 16

L. 88. ‘(in general - measured in watts per square meter, ?/?2)’ – it is worth considering inclusion of more
specific information (range of flux values for A, B, C, M, X flares) once physical units are given.
Autors Response: we understand our omission, and a classification with specific values with watts per square meter for each eruption class is already given, the above mentioned sentence now is: "In general solar flares are measuren in watts per square meter - 〖W/m〗^2, and their are clasiificaed as A (〖〖<10〗^(-7) W/m〗^2,), B (〖〖10〗^(-7) 〖÷10〗^(-6) W/m〗^2), C (〖〖10〗^(-6) 〖÷10〗^(-5) W/m〗^2), M (〖〖10〗^(-5) 〖÷10〗^(-4) W/m〗^2) and X (〖〖>10〗^(-4) W/m〗^2)"; now is  line 87-89    

L. 99. ‘ions carriers’ – what are ‘ions carriers’ in the ionosphere
Autors Response:  sentence now is :"However, the concentration of ions propagandation in the ionosphere also depends on several additional atmospheric meteorological factors, which introduces some uncertainty in the prediction of solar flares." We think the propagandation is more relevante than carrier; now is line 100-102

L. 207. ‘on Figure 2’ – should be ‘in Figure 2’.
Autors Response: fixed like recomendation, now is 208

L. 208. ‘API’ – the abbreviation should be explained, ‘Application Programming Interface’.
Autors Response: fxed like recomendation now in brackets is given API (Application Programming Interface); now is line 209

L. 216. ‘10kHz - 30MHz’ – spaces are desired before the units, ’10 kHz – 30 MHz’.
Autors Response: fxed like recomendationр ; now is line 2018

L. 219. ‘adjust the receive bandwidth’ – ‘receive’ is not a noun, ‘receiver’ or ‘reception’ should be.
Autors Response:fxed like recomendation, now a noun reciver is used; line 221 

L. 247. ‘Round-Robin Database (RRD)’ – this abbreviation is not used in the text and may be eliminated
Autors Response: the abbreviation (RRD) is removed; now is lineline 249

L. 292 (scheme). ‘Is the information time>lenght’ – should be ‘length’ instead of ‘lenght’.
Autors Response:fxed like recomendation; now is line line 294

L. 321. ‘The data collected by the developed system are compared with published and measured results
concerning solar activity by other means’ – should be edited, the meaning of the phrase ‘by other means’ is
unclear. What is considered as published results and what does as measured ones?
Autors Response: the whole sentence has been rewritten and now reads as follows "The data collected by the developed system are compared with published and measured results concerning solar activity by measurements using detection technology (like publick data from SpaceWatherlive, referred to in this point) and relevant conclusions are drawn concerning the feasibility of development."; now is line 322- 325

L. 348. ‘The information on Table 1 shows that the days with geomagnetic storms G1 and G2 in the selected
period are 26.07.2023 and 05.08.2023’ – an incomprehensible text; Table 1 applies to input system parameters,
not days with geomagnetic storms. The information should be verified because NOAA estimated geomagnetic
storm on 5 August 2023 as G3 (https://www.swpc.noaa.gov/news/strong-g3-geomagnetic-storms-observed-05-
aug-2023).
Autors Response: we understand the ambiguity arising on reading, the remark is relevant, the whole sentence has been rewritten and now has the form "The information propoused from NOAA (https://www.swpc.noaa.gov/) shows that in the selected period are 26.07.2023 and 05.08.2023 had geomagnetic storms G1, G2 and G3. This is a major proof point based on actual confirmed data that the system can be used to identify such events."; now is line 351-355

L. 354. ‘M4.5 flare’ – the flare is indicated in Fig. 7 as M4.63.
Autors Response:fxed like recomendation; now is line line 358
L. 392. ‘their recombination in 3’ – what does ‘in 3’ mean?
L. 394. ‘This indicates that the measurement interval of the system should be infused at an interval not greater
Regarding the comments in the third chapter, the authors changed the presentation of the results and presented data from technologies working with another principle for the detection of solar flares, for example, a satellite system for direct measurement of solar flares GOES. Chapter Three has been completely changed

L. 449. ‘ “Conceptualization’ – the opening quote mark should be removed.
Autors Response:fxed like recomendation;
L. 450. ‘Y.V..’ – the double dot should be eliminated.
Autors Response:fxed like recomendation; now is line line 454

L. 456. ‘Recov-ery’ – the hyphen is unnecessary.
Autors Response:fxed like recomendation; 
L. 483. ‘ground waves’ – original form in the article of King and Maley is ‘groundwaves’    
Autors Response:fxed like recomendation; 

L. 490. ‘fare’ – should be ‘flare’.
Autors Response:fxed like recomendation; 

Reviewer 2 Report

Comments and Suggestions for Authors

(1) The reply [2] about the abstract is not acceptable, as the author claims that "not necessary to build independent measuring stations, but a ready-made network of SDRs can be used", meaning the SDR network can independently determine solar activities, then the author needs to do a direct comparison of between two to prove this, and provide statistic numbers (the sensitivity, smallest detectable flare level, the time difference between NOAA detected flare and author proposed method... etc).

(2) the claim of "and the ability to determine solar activity is assessed" need to be supported by numbers.

(3) the plot should be improved, I suggest referring to the style of the following plot. Now there are the following problems: no y-label, time-axis have very redundant components (year day month is the same, +00:00 is UT can also go into x-label instead of in every x-tick), the plot from the same day can be merged into one (different station use different color), this can provide a more direct comparison.

(4) The conclusion still has the problem of no supportive numbers and statistics, measurements, sensitivity, and time differences of flare and ionosphere disturbances.

Comments on the Quality of English Language

Needs to be improved

Author Response

1) The reply [2] about the abstract is not acceptable, as the author claims that "not necessary to build independent measuring stations, but a ready-made network of SDRs can be used", meaning the SDR network can independently determine solar activities, then the author needs to do a direct comparison of between two to prove this, and provide statistic numbers (the sensitivity, smallest detectable flare level, the time difference between NOAA detected flare and author proposed method... etc).
Author's reply: The authors note that it is not necessary to smooth an independent base station, but it is possible to use a ready-made network of receivers

(2) the claim of "and the ability to determine solar activity is assessed" need to be supported by numbers.
Author's reply: data from correlation and comparisons have been added, chapter three has been expanded

(3) the plot should be improved, I suggest referring to the style of the following plot. Now there are the following problems: no y-label, time-axis have very redundant components (year day month is the same, +00:00 is UT can also go into x-label instead of in every x-tick), the plot from the same day can be merged into one (different station use different color), this can provide a more direct comparison.
Author's reply: The graphics have been improved, and we have taken the reviewer's recommendations into account when making the change

(4) The conclusion still has the problem of no supportive numbers and statistics, measurements, sensitivity, and time differences of flare and ionosphere disturbances.
Author's reply:  Added comparison and correlations between Super Sid Irish Indirect Solar Activity Observation System and comparison with data from direct measurement of solar flares from two GOAS satellites

Reviewer 3 Report

Comments and Suggestions for Authors

The authors have corrected most of the bugs.

If the publisher's policy allows borrowing from the manual, then you can leave Fig. 3.

The timestamps in Fig. 10 are bad. Even the author himself uses only hours and minutes in the text. The day in the example shown does not change. Why such a label format and so many decimal places? The record in the data file can be anything, but it is unacceptable to use such labels in the article. If the authors do not want to combine all these figures into a stack with a common time axis, then let them redo the tick labels as shown in the attached file.

The scientific part of the article is still weak, but the obvious shortcomings have been removed. You can publish after editing the pictures.

Author Response

Article three was completely redone, which also led to an increase in the quality of the figures, the proposal to collect the figures in a common coordinate system was taken into account. But this is not the only thing that has been done here, a comparison and data of correlation of obtained data with other systems for detection of solar flares - direct and indirect system has been added, the obtained data is supported by numbers. We hope that with the changed results and data, the scientific part has been improved. We want to remind once again that our single claim is for a proposed selected and tested approach and not for a new method.